# A Multi-Pollutant and Meteorological Analysis of Cardiorespiratory Mortality among the Elderly in São Paulo, Brazil—An Artificial Neural Networks Approach

**DOI:** 10.3390/ijerph20085458

**Published:** 2023-04-11

**Authors:** Luciana Leirião, Michelle de Oliveira, Tiago Martins, Simone Miraglia

**Affiliations:** Institute of Environmental, Chemical and Pharmaceutical Sciences, Federal University of Sao Paulo (UNIFESP), Diadema 09913030, Brazil

**Keywords:** air pollution, health effects, multi-pollutant model, artificial neural network

## Abstract

Traditionally, studies that associate air pollution with health effects relate individual pollutants to outcomes such as mortality or hospital admissions. However, models capable of analyzing the effects resulting from the atmosphere mixture are demanded. In this study, multilayer perceptron neural networks were evaluated to associate PM_10_, NO_2_, and SO_2_ concentrations, temperature, wind speed, and relative air humidity with cardiorespiratory mortality among the elderly in São Paulo, Brazil. Daily data from 2007 to 2019 were considered and different numbers of neurons on the hidden layer, algorithms, and a combination of activation functions were tested. The best-fitted artificial neural network (ANN) resulted in a MAPE equal to 13.46%. When individual season data were analyzed, the MAPE decreased to 11%. The most influential variables in cardiorespiratory mortality among the elderly were PM_10_ and NO_2_ concentrations. The relative humidity variable is more important during the dry season, and temperature is more important during the rainy season. The models were not subjected to the multicollinearity issue as with classical regression models. The use of ANNs to relate air quality to health outcomes is still very incipient, and this work highlights that it is a powerful tool that should be further explored.

## 1. Introduction

Air pollution is the first environmental risk factor for the world’s population [1]. According to the Global Burden of Disease (GBD), in 2019, there were 6.67 million excess deaths due to exposure to particulate matter pollutants [2]. Other authors argue that the world’s mortality due to particulate matter exposure may be even greater if other death causes behind the ones considered by the GBD were computed. Burnett et al. [3], for instance, calculate that particulate matter with a diameter of 2.5 µm or less (PM_2.5_) could have been responsible for 8.5 million deaths in 2015 worldwide.

The literature concerning the harmful effects of air pollution exposure has been increasing since 2011 [4]. Although the majority of the studies associate the pollutant concentrations with respiratory diseases, chronic diseases, and cardiovascular diseases [5,6,7,8], recent research has emerged suggesting that air pollution may be associated with mental disorders and other diseases, such as DNA methylation changes, inflammatory disease, skin disease, and abortion [4,9].

The effects of air pollution are not the same across the population; the elderly, children, and people with chronic diseases are the main risk group [4]. Studies point to higher risks in the elderly compared to the rest of the population, probably as a consequence of pre-existing disease complications, and the main pollutant affecting this group is particulate matter [10,11,12]. Temperature and relative humidity increases also appear to play a fundamental role in mortality [13,14,15].

Several authors around the world have aimed to understand the health effects among the elderly of an increase of 10 µg/m^3^ or 50 µg/m^3^ in the daily concentration of pollutants such as particulate matter (PM) and sulfur dioxide (SO_2_). The most discussed outcomes are hospitalizations related to the respiratory system or chronic obstructive pulmonary disease and deaths from respiratory causes [16,17,18,19,20,21]. Studies aiming to associate elderly mortality with pollutant exposure are mainly single-pollutant analyses. In order to better understand the role of air pollution in the elderly, more investigations focusing on exposure to multi-pollutant mixtures are needed [11].

Despite the wide variety of studies around the world, they are mainly concentrated in countries with a temperate climate, being less frequent in countries with a tropical climate [22]. Among tropical countries, Brazil stands out as one of the most studied countries (along with India and Australia) [22]. Studying air pollution health effects jointly with temperature variation is important in these countries since the results found may be quite different from the ones obtained in temperate countries, given the lower temperature variation throughout the year [23].

Among the regions with a tropical climate, the municipality of São Paulo (Brazil) stands out in relation to studies that aim to estimate the health effects of air pollution exposure. The city is among the 10 most populous in the world, being the largest in Latin America with more than 12 million inhabitants [24]. According to a 2016 study, which so far is the most recent application of the official WHO methodology in São Paulo, about 5000 premature deaths annually can be associated with excess PM_2.5_ in the atmosphere [25]. Vehicles are mainly responsible for the high concentration of pollutants in the atmosphere, and the official estimation is that there are almost 9 million vehicles in the municipality (8% of the national fleet in 0.17% of Brazil’s territory) [26]. The city’s status as a global economic hub (31% of Brazil’s GDP [27]) attracts people with a typical profile that prefers using their car to public transportation. Thus, the emission of air pollutants and the large number of people exposed to these pollutants position São Paulo as a global challenge in terms of its environmental and public health issues.

Since the 1990s, several studies relating adverse health effects to pollutant concentrations have been carried out in São Paulo [28,29,30,31]. The most recent studies (from 2015 on) considered the particulate matter with a diameter of less than 10 µm (PM_10_), nitrogen dioxide (NO_2_), carbon monoxide (CO), ozone (O_3_), and SO_2_ pollutants, and several health outcomes. Bravo [32] investigated the relationship between PM_10_, NO_2_, SO_2_, CO, and O_3_ (individually analyzed) and non-accidental, cardiovascular, and respiratory deaths among city residents and found a positive relationship for all analyses, except between O_3_ concentration and cardiovascular deaths. Abe et al. [33] demonstrated a positive relationship between PM_10_ concentration and cardiovascular and respiratory deaths. Finally, Santana et al. [34] demonstrated the positive relationship between pollutants PM, O_3_, CO, SO_2_, and NO_2_ (individually analyzed) and hospitalizations due to respiratory diseases. 

A few recent studies have focused on the health effects among the elderly population in São Paulo. Costa et al. [35] found that a 10 µg/m^3^ addition to PM_10_ and NO_2_ concentrations increased nonaccidental deaths among the elderly by 0.35% and 0.40%, respectively. According to the authors, the pollutant effects are even higher two days after the exposure (0.6% and 0.58% for PM_10_ and NO_2_, respectively) [35]. Ferreira et al. [36] specifically analyzed the effect of particulate matter on hospital admissions among the elderly and concluded that the ionic composition of the pollutant (specifically the presence of SO_4_^2−^ in PM_10_ and K^+^ in PM_2.5_) increases the health effects in this population. Finally, it is also highlighted that the elderly population of lower socioeconomic levels seems to be more affected in relation to air pollution in São Paulo [37].

Supported by scientific and technological development, São Paulo and several governments around the world have implemented actions aiming to reduce the concentration of pollutants in the atmosphere. These policies are mainly focused on particulate matter, which is the pollutant most frequently associated with health problems. However, the focus on single pollutants neglects the fact that the atmosphere is a complex mixture of substances subject to the action of physical and chemical processes and that individuals are not exposed to a single pollutant at a time but simultaneously to a mixture of them [38].

Scientific development aiming at understanding the joint effect of pollutants on human health has resulted in a series of models called multipollutant models (as opposed to the ones that consider only one pollutant at a time, called single pollutant models). One of the first studies of this type was carried out by Gold et al., in 1999, which considered the cumulative effect of PM_2.5_ and O_3_ on lung function [39]. According to Davalos et al. [40], the main models that have been adopted to analyze the joint effect of exposure to pollutants are additive main effects (AMEs); effect measure modification (EMM); unsupervised dimension reduction (UDR); supervised dimension reduction (SDR); and nonparametric methods. The lack of consensus on a specific model for this analysis is due to statistical challenges, such as the multicollinearity of the independent variables, and the interaction among components of the atmosphere [41].

The pollutants usually considered in multipollutant models are particulate matter (PM_2.5_ or PM_10_), nitrogen oxides (NO_x_), and ozone (O_3_) [40]. The health effects of exposure can be investigated by considering the pollutants simultaneously or in pairs. The results of several studies on the interaction of pollutants and their health effects are contradictory. While some authors describe additive effects, others describe independent or even antagonistic effects. In Colombia, Rodríguez-Villamizar et al. [42], through conditional negative binomial regression models, found an additive effect of NO_2_ in relation to PM_2.5_ when analyzing their association with cardiovascular hospitalizations. A similar result was found by Jiang et al. [43] in China regarding the addition of PM_2.5_ to the O_3_ effect. In this case, the authors used a generalized linear Poisson model. On the other hand, using the same statistical model, Zhang et al. [44] found a decrease in the association between NO_2_ and cardiovascular mortality when the model was adjusted considering SO_2_ and PM_10_ pollutants. More recently, Shin et al. [45] applied generalized additive over-dispersion Poisson regression models and described that the individual effects of O_3_, NO_2_, and PM_2.5_ pollutants were not considered under or over-estimated when compared with multi-two or three pollutants models.

Health outcomes, in general, constitute a multifactorial problem. Thus, there is no medical consensus on the total number of variables and their importance in relation to these outcomes. The interaction between several variables (such as pollutant exposure, meteorological conditions, gender, life habits, age, etc.) is still unknown to medicine and this makes it difficult to explain the physiology of the phenomenon. Considering the statistical difficulty regarding the traditional models, artificial neural networks (ANNs) emerge as a powerful tool for this type of study. An ANN can manage a large complexity of information without the need to know the phenomenon itself (whether or not the variables are correlated). When interactions are so complex that it is difficult to separate one effect from another, ANNs can be a more interesting mathematical solution than simpler mathematical models that consider only one or a smaller number of variables.

ANNs are learning algorithms inspired by the nervous system of superior organisms [46]. So, they are based on simple processing units (neurons), which store experimental knowledge and make it available for use through a learning process [46]. In ANNs, neurons are distributed in one or more interrelated layers. Based on the existing data, the neurons are trained to estimate outputs from the input data. ANNs have been widely applied in air quality forecasting (mainly concerning particulate matter, nitrogen oxides, and ozone) based on meteorological variables, emissions, and traffic data [47,48]. Since they deal in a non-linear way with a large volume of data and have a high capacity to make generalizations, ANNs have a high potential to be explored to estimate health outcomes from air pollutant exposure [49].

São Paulo stands out in terms of the number of publications relating air pollution to health effects [22]. Although ANNs are still little applied in these studies around the world, some authors have used this approach in São Paulo. This allows comparisons and a margin for improvement. The comparison between the classical generalized linear model (GLM) and the artificial neural network in São Paulo revealed that the ANN shows better results when modeling the daily hospital admissions due to PM_10_ exposure and meteorological conditions [49]. In a multipollutant analysis, Miranda et al. [50] considered the monthly mean concentration of six air pollutants (PM_10_, PM_2.5_, O_3_, CO, SO_2_, and NO_2_) to estimate hospitalizations for respiratory diseases. The best ANN found by the authors was able to estimate 85.5% of the experimental data, and the percentage errors (from 2% to 25%) were considered low [50]. Despite the good result, the study did not consider meteorological variables. In all previous studies, the best-fitted ANN had difficulties in estimating the highest and lowest values of the time series.

Bearing all this in mind and considering the difficulties and limitations inherent to the most traditional multipollutant models, this study aimed to use multilayer perceptron artificial neural networks to find a model able to associate PM_10_, NO_2_, and SO_2_ concentrations and meteorological variables (temperature, wind speed, and relative air humidity) with cardiorespiratory mortality among the elderly population in São Paulo, Brazil.

## 2. Materials and Methods

### 2.1. Study Area, Data Sources, and Processing

The municipality of São Paulo is located in the Southeast region of Brazil, extends over 1521 km^2^, and is estimated to have 12.3 million inhabitants [24]. Air quality is monitored by 18 monitoring stations controlled by the environmental company of the State of São Paulo (named CETESB—Companhia Ambiental do Estado de São Paulo), which are heterogeneously distributed (Figure 1). In 2019, the annual average concentration of PM_10_, PM_2.5_, and NO_2_ pollutants in the municipality was, respectively, 28.66 µg/m^3^, 16.41 µg/m^3^, and 35.66 µg/m^3^ [51]. These values were above the annual averages recommended by the World Health Organization, which are, respectively, 15 µg/m^3^, 5 µg/m^3^, and 10 µg/m^3^ [52]. Even during 2020, when several services were paralyzed due to the COVID-19 pandemic, the annual concentrations of these pollutants were well above recommended levels (27.25 µg/m^3^, 14.78 µg/m^3^, and 30.77 µg/m^3^ for PM_10_, PM_2.5_, and NO_2_, respectively) [53].

To obtain the ANN proposed in this study, hourly air quality data were obtained from the CETESB database in all 18 monitoring stations from 1 January 2007 to 31 December 2019 (all data can be consulted on https://qualar.cetesb.sp.gov.br/qualar/home.do (accessed on 20 August 2022)). The data considered the concentration of PM_10_, PM_2.5_, O_3_, NO_2_, and SO_2_ pollutants. Due to the existence of missing data in the database, PM_2.5_ and O_3_ data were disregarded. For the other pollutants, daily mean concentration was calculated considering only days with more than 75% of hourly data available. 

For the calculation of the daily average concentration of the pollutants in the city of São Paulo, only the monitoring stations that had data for the entire time series were considered (Figure 1). So, the selection of the monitoring station respected only one criterion, which was the availability data for the entire time series. In relation to PM_10_, the stations considered were: Cerqueira César, Congonhas, Grajaú, Nossa Senhora do Ó, and Parque D. Pedro II. For NO_2_, the stations Cerqueira César, Congonhas, Ibirapuera, Parque D Pedro II, and Pinheiros were considered. Finally, for SO_2_, the stations considered were Cerqueira Cesar and Congonhas.

The meteorological data considered in the investigation were daily means of air temperature, relative humidity, and wind speed. They were obtained from the Institute of Astronomy, Geophysics and Atmospheric Sciences of Sao Paulo University (IAG-Instituto de Astronomia, Geofísica e Ciências Atmosféricas da Universidade de São Paulo), which is located in the southern area of Sao Paulo municipality. The data are provided by completing a form on their website (http://www.estacao.iag.usp.br/sol_dados.php (accessed on 20 August 2022)).

Daily mortality due to cardiorespiratory disease (chapters IX and X of ICD) data were obtained from the Informatics Department of the Brazilian Health System (named DATASUS). The data are open access and can be consulted on DATASUS website (https://datasus.saude.gov.br/informacoes-de-saude-tabnet/ (accessed on 20 August 2022)). The database contained all-age mortality and was filtered to consider only the deaths among the population older than 60 years. As the elderly population in the city grew by 36% during the analyzed period (from 1.2 million in 2007 to 1.9 million in 2019), the number of deaths was relativized by the total elderly population (>60 years) of each year. The population data for this relativization was consulted from the State of Sao Paulo System of Data Analysis (named Fundação SEADE) on the website https://municipios.seade.gov.br/ (accessed on 20 August 2022).

### 2.2. Artificial Neural Network

Using MATLAB programming language, several multilayer perceptron (MLP) neural networks were designed. The daily concentration of the three pollutants and meteorological variables were considered as inputs, and the respective cardiorespiratory mortality among the elderly population was considered as output (Figure 2).

On MLPs, each neuron receives a synaptic weight, which represents the relative influence of the input data on the respective neuron. The result of the multiplication between the variable and its weight summed with a bias parameter results in an activation coefficient, as follows (Equation (1)):(1)$$\alpha_{j}=\sum_{i=1}^{z}w_{ij}x_{i}+b_{j}$$
where α*_j_* is the activation value; *z* is the number of inputs; *x_i_* is the neuron’s input variable *i*; *w_ij_* is the weight of the variable *i* on the neuron *j*; and *b_j_* is the bias [46]. The neuron’s output is then calculated by using an activation function over *α_j_*, which can be linear or non-linear functions (such as hyperbolic tangent and sigmoidal).

The MLP uses an optimization algorithm to minimize an objective function (OF) that is related to the difference between the obtained output value and the expected one. This minimization is performed through a supervised learning technique to determine the weight and the bias of each neuron [46].

In this study, the MLP neural networks with one and two hidden layers were tested. Structures with 5, 10, 15, 20, 25, 30, and 35 neurons in the hidden layer were considered. More neurons were not used, because it would generate an overfitted model. Four different learning algorithms were used (Scaled Conjugate Gradient-trainscg, Levenberg–Marquardt-trainlm, Powell–Beale conjugate gradient backpropagation-traincgb, and Bayesinan Regularization-trainbr), and the combinations of three activation functions (hyperbolic tangent sigmoid-tansig, linear-purelin, and logarithmic sigmoid-logsig) were tested. As we showed in previous studies, testing different learning algorithms, and the random separation of training and validation data, are important to achieve a good result. The main reason for including several ANN configurations was to increase the search for the configuration that leads to a good result. It totalized 420 ANN architectures. Each ANN configuration was trained 50 times. From the total obtained data, 70% was used for training, 15% for validation, and the remaining 15% for the test. Data division was randomly performed by the software. The validation step is performed at the same time as the training step, to prevent overfitting. The validation objective function is monitored during training, and the one stop criterion is the increase in this parameter when the training data objective function continues to decrease. Test data are used for an independent prediction after the training stopped.

The mean square error (MSE) was the objective function to be minimized. The errors of each of these steps were considered, and the network with the lower mean absolute percentage error (MAPE) was considered as the one with the best fit.

After determining the best ANN model, the importance (weight) of each variable to the cardiorespiratory mortality was calculated according to connection weight method [54]. For each variable *i*, we calculated the variable *I_ij_* as follows (Equation (2)):(2)$$I_{ij}=w_{ij}⬝w_{1j}$$
where *w_ij_* is the weight matrix of the hidden layer, being *i* input variables and *j* neurons in the hidden layer, and *w*_1*j*_ is the weight matrix of the output layer, being 1 output and *j* inputs (from the hidden layer). So, the importance of the input *i* (*Imp_i_*) could be obtained by Equation (3):(3)$$Imp_{i}=\sum_{j}I_{ij}$$

The seasonality behind all considered variables is widely described, with increases in pollutant concentrations and mortality during the dry season (from April to September) and reductions in the rainy season (from October to March) [53]. This characteristic hampers the neural networks from accurately estimating the highest and lowest values, as detected by Araujo et al. [49] and Miranda et al. [50]. For this reason, in this study, three models were developed. The first one (model I) considered all the collected data (4748 daily observations collected from 2007 to 2019). The second one (model II) considered only data from the rainy season (2369 daily observations considering only the months from October to March of the same year interval). Finally, the third model (model III) considered the data from the dry season (2379 daily observations considering only the months from April to September of the same year interval).

### 2.3. Generalized Linear Model

As the ANN approach is still little explored in investigating the relationship between air pollution or meteorological variables and health outcomes, the ANN models were validated by comparing the results with a traditional approach. Generalized linear models (GLMs) were originally defined by Nelder and Wedderburn in 1972 [55] as an extension of conventional linear models. GLMs are capable of dealing with distributions of the response variable different from the normal distribution and the relationship between input variables and output variables different from the linear [56].

In this study, a GLM with Gamma distribution was adopted due to the nature of the response variable (positive and continuous numbers). The link function adopted was of a loglink, resulting in the model represented by Equation (4):(4)$$\mu_{i}=exp(\beta_{0}+\sum_{1}\beta_{1}x_{i})$$
where *µ_i_* is the response variable; *β* is estimated by the model, and *x_i_* is the independent variable.

Three models were developed using SPSS software. The first model (Model I’) considered the entire database. The second (Model II’) considered only the observations (daily measures) that occurred in the rainy season. Finally, the third (Model III’) considered only the observations that occurred in the dry season.

As suggested by Conceição et al., for the construction of GLM models, a basic model was first built before adding the concentrations of pollutants [56]. In this basic model, the following explanatory variables were considered: month, year, day of the week, and meteorological variables (relative humidity, temperature and wind speed). Once the models with the best set of controls for seasonality and climate were defined, pollutant concentrations were included. So, Models I’, II’, and III’ had as input variables the daily concentrations of PM_10_, NO_2_, and SO_2_, and the variables considered as significative (*p* < 0.05) in the basic models. As response variables, we adopted cardiorespiratory mortality among the elderly (deaths per 100,000 elderly inhabitants).

The models were applied to estimate the response variable in all observations, and the MAPE was estimated. The distributions of MAPEs in the three models (Model I’, Model II’, and Model III’) were compared with the distributions of MAPEs resulting from neural networks (Model I, Model II, and Model III).

## 3. Results

### 3.1. Descriptive Statistics

Over the years, there has been a decrease in the average concentration of the three pollutants considered in this study. The reduction was approximately 31.6% for PM_10_ (from 41.49 µg/m^3^ in 2007 to 28.35 µg/m^3^ in 2019), 31.3% for NO_2_ (from 58.36 µg/m^3^ to 40.08 µg/m^3^), and 82.5% for SO_2_ (from 9.66 µg/m^3^ to 1.68 µg/m^3^) (Figure 3).

Regarding the meteorological variables, there were few changes over the analyzed period, except for the air relative humidity in 2014, which was considered an extremely dry year in the region of São Paulo [57]. It is observed that, in that year, the relative humidity was the lowest in the time series and that, during the rainy season, it was even lower than the one recorded during the dry season (76.5% during the rainy season versus 78.9% during the dry season) (Figure 4).

As well as the pollutants’ concentrations, the daily mortality rate due to cardiorespiratory problems in the elderly decreased between 2007 and 2019 (from 5.2 to 4.08 per 100,000 inhabitants) (Figure 3). In both the pollutant concentrations and the cardiorespiratory mortality, is possible to clearly note the seasonality of the data, with peaks during the dry season (Figure 3).

### 3.2. Artificial Neural Networks

By analyzing the results of Model I (which considered all the available data for rainy and dry seasons), the ANN that best fitted the data had only one hidden layer and was composed of six input neurons, 25 neurons on the hidden layer, and one output neuron. The learning algorithm was trainlm, and the combination of activation functions was tansig and purelin. The value of the objective function was 0.614 for the training step, which is very low. The mean absolute percentage error (MAPE) of the ANN configuration was 13.46%. The mean errors of all ANN configurations ranged from 13.46% to 54.93%.

Based on this ANN, the estimated cardiorespiratory mortality (deaths per 10^5^ elderly inhabitants) ranged between 0.59 and 6.29 while the real mortality ranged from 2.03 to 8.58. The errors of these estimates ranged between 0% and 131%. The biggest errors were concentrated in the estimates of the highest values of cardiorespiratory mortality (Figure 5).

Considering Model II, in which only the data from the rainy season were used, the best ANN was the one hidden layer ANN composed of six input neurons, 30 neurons on the hidden layer, and one output neuron. The learning algorithm and the combination of activation functions were the same as the best ANN, considering the entire database (learning algorithm: trainlm; activation functions: tansig and purelin). The value of the objective function in the training step was equal to 0.375. and the MAPE was equal to 11.68%. The mean error of all ANN configurations ranged from 11.68% to 81.53%.

Based on this ANN, the estimated daily cardiorespiratory mortality ranged between 3.43 and 6.23 deaths per 10^5^ elderly inhabitants. The real mortality considering only the data from the rainy season ranged from 2.30 to 8.09. The errors of these estimates ranged between 0% and 79% (Figure 6).

Finally, Model III, in which only data from the dry season were considered, the best-fitted ANN also had only one hidden layer. Its structure was six input neurons, 30 neurons in the hidden layer, and one output neuron. The learning algorithm was also trainlm, and the combination of activation functions, tansig and purelin. The value of the objective function of the training step was 0.452. The resulting mean absolute percentage was 11.07%. The mean error of all ANN configurations ranged from 11.07% to 58.55%.

Based on this ANN, the estimated cardiorespiratory mortality ranged between 3.64 and 8.60 deaths per 10^5^ elderly inhabitants while the observed mortality ranged from 2.03 to 8.58. The errors of these estimates ranged between 0% and 144%. As observed in the results of the best-fitted ANN considering the entire database, the biggest errors were concerned with estimates of the lowest values of cardiorespiratory mortality (Figure 7).

The best structures of each one of the models were tested through 50 training iterations. The MSE distribution test is presented in Figure 8. For the models I, II, and III presented in the study, the MSE were 0.616, 0.375, and 0.452, respectively. The dispersion of the MSE presented a similar range for the three models (between 0.12 and 0.14).

The three models presented errors lower than 20% in more than 75% of the observations (Figure 9). Model I (considering the entire database) was the one with the highest MAPE (13%). By dividing the database according to season (rainy or dry), the MAPE was reduced to 11% in both models (I and II). This mean reduction was considered statistically significant (comparison between Models I and II: *t* test (7115) = 6.428; *p* < 0.05. Comparison between Models I and III: *t* test (7115) = 9.311; *p* < 0.05). The adoption of different models according to the season was also reflected in the reduction in outliers in the forecasts (Figure 9).

The structures of the best-fitted ANNs described above can be represented according to Equations (5) and (6):

Model I:(5)$$y_{k}=\sum_{j=1}^{25}w_{jk}\mathrm{tansig}{\left(\sum_{i=1}^{6}w_{ij}x_{i}+b_{j}\right)}_{j}+b_{k}$$

Models II and III:(6)$$y_{k}=\sum_{j=1}^{30}w_{jk}\mathrm{tansig}{\left(\sum_{i=1}^{6}w_{ij}x_{i}+b_{j}\right)}_{j}+b_{k}$$
where *y_k_* is the desired output, *w_ij_* and *w_jk_* are the weights of the neurons in the hidden and output layers, respectively, and *b_j_* and *b_k_* are the bias of the neurons in the hidden and output layers, respectively. The weights and the bias for each model are given in the Appendix A.

After defining the best ANN for the three considered datasets, the determination of the influence of the inputs was obtained by using the connection weight method. The results are shown in Table 1. Among all the considered variables, for Model I, the results showed that SO_2_ concentration was considered the most important variable, with 46.75% weight. When the rainy (Model II) and dry (Model III) seasons were separately analyzed, the PM_10_ concentration was revealed as the most important variable for both ANNs, with the importance of 68.34% and 29.07%, respectively. NO_2_ concentration was presented as the second or third most influential variable in the models. Concerning the meteorological variables, wind speed is the least important variable in all three models. Temperature is more important during the rainy season, which comprises the summer (higher mean temperature), and relative humidity is more important during the dry season when the mean relative humidity is lower.

### 3.3. ANN Validation by the Comparison with GLM Model

According to the basic model results, the whole set of variables was considered in Models I’ and III’, and only the wind speed was excluded from Model II’(*p* = 0.125).

In Model I’, considering the entire database, all input variables except for PM_10_ concentration, temperature, and wind speed were considered relevant to cardiorespiratory mortality among the elderly (*p* < 0.000). As in the analysis of importance performed from Model I, in Model I’, the concentration of SO_2_ showed a greater impact on the response variable than the concentration of the other pollutants analyzed (*β* = −0.003). Wind speed was excluded from Model II’ because it presented a *p*-value > 0.05 in the previous basic model. All other variables were considered significative in Model II’. In Model III’, wind speed was the only variable that did not present a *p*-value < 0.05. The parameter estimates for the pollutants in Models II’ and III’ were very similar to the ones estimated in Model I’ (Table 2).

The separation of the database into two according to the season resulted in lower MAPEs for both Model II’ (rainy season; *t* test (7115) = 4.323; *p* < 0.000) and Model III’ (dry season; *t* test (7115) = 5.311; *p* < 0.000). Model I’ presented a MAPE equal to 12.49%. and in Models II’ and III’, the MAPEs were equal to 11.08% and 10.76%, respectively. The MAPE distribution on each one of the models was very similar to the distribution resulting from the ANN models (Figure 10).

## 4. Discussion

Air quality improvement in the city of São Paulo has been consistently observed over the last 30 years as a result of a series of environmental policies [58]. Throughout the 1990s, the city experienced an exit from industries, which settled in nearby regions in the interior of the state, and its economy began to be based on the services sector and business management [59]. As a consequence of industrial migration and the increase in the number of inhabitants from medium and high-income groups, mobile sources are the main factors responsible for air pollution [53,58]. Therefore, the main policies for the improvement in air quality observed since the 2000s are the no-drive day (in force since 1997) and a national program that imposes new technologies on vehicles and fuels, named PROCONVE [60,61]. Regarding PROCONVE, the mandatory use of catalysts in engines and the reduction in fuel sulfur content were responsible for the reduction in the CO, hydrocarbons, and SO_2_ atmospheric concentrations [62,63,64,65]. In addition to these programs, more specific measures aimed at reducing heavy-duty vehicle emissions were adopted, such as the construction of a beltway road and driving restrictions for trucks in some corridors during the day time. These measures have contributed to reducing the concentration of pollutants such as NO_x_ and PM, bringing health improvements [66,67]. Despite the positive results attributed to these environmental policies, São Paulo continues to have air pollution levels above those recommended by the WHO, representing a constant risk to the health of the population [58]. So, studies aiming to model the health impacts of the air quality in Sao Paulo are still of great importance even with the reduction in the concentration of pollutants.

The seasonality observed in the concentration of pollutants is also followed by a seasonality in cardiorespiratory mortality among the elderly (see Figure 3). The annual variation in the concentration of pollutants is a consequence of the meteorological characteristics typically found in the two seasons of the year (rainy and dry), with the months between May and September being the most unfavorable to the pollutants’ dispersion [53]. According to Sánchez-Ccoyllo and de Andrade [68], there is an increase in the concentration of particulate matter in the Metropolitan Region of São Paulo on days with high temperature, low relative humidity, absence of precipitation, and weak ventilation. Considering this, the worst pollution days tend to occur during the winter (between June and September), when these factors are more frequently associated [65]. In winter periods, the phenomenon of thermal inversion is also common in the municipality, which makes it difficult to disperse pollutants [69]. Air pollution is worse in the winter because colder and drier air traps more pollution. Added to this, long periods of drought in this season lead to occasional increases in the concentration, mainly of particulate matter [53]. Possibly, meteorological conditions are mainly responsible for the difference in the concentration of pollutants throughout the year, since it has already been demonstrated that the origin of particulate pollutants remains very similar during summer and winter [70]. In São Paulo, the winter is characterized by the driest period of the year. Thus, in addition to the greater amount of pollutants that penetrate the respiratory system, in this season, the elderly are also more susceptible to infections such as influenza and pneumonia, which are common during the winter and contribute to cardiorespiratory impairment [71,72]. The air pollution increment and the increased occurrence of respiratory diseases during the dry season explain the seasonality observed over the analyzed time series.

Unlike the models most frequently used to relate meteorological and pollution data with health outcomes, ANNs do not need prior adjustments due to the interaction between the input variables [73]. ANN models also do not require prior adjustments due to the seasonality of the data. In the application of the GLM, for example, the smoothing of seasonality must be undertaken considering explanatory variables related to this seasonality in the model or through techniques such as loess and cubic splines [56]. Thus, this work could simultaneously assess the effect of six variables on cardiorespiratory mortality among elderly people in São Paulo without previous data treatment. This possibility represents a great advance in relation to the linear models since these classical models are not efficient in managing interactions among the input variables [38]. In addition to the ability to manage multicollinearity, several studies that compared ANNs with other approaches pointed out that ANNs result in models with better precision [49,74,75].

By allowing the analysis of several variables regardless of the relationship among them, studies that apply ANNs to estimate the effects of air pollution on human health vary greatly in relation to the explanatory variables. Some studies, for instance, considered only pollutant concentrations [75,76]. Other studies included meteorological variables, days of the week, and traffic data as explanatory variables as well [77,78]. The presentation of the model’s quality also varies among the studies, with indicators such as mean square error (MSE), mean error (ME), mean absolute percentage error (MAPE), and R-square (R^2^). The diversity of variables and results constitutes an obstacle to the comparison of the results obtained in the studies.

Previous ANNs that related human health outcomes to pollutant concentrations in the São Paulo municipality reached higher MAPEs than the ones found in this work. Araujo et al. [49] considered PM_10_ concentration, temperature, relative humidity, days of the week, and holidays to estimate respiratory hospital admissions. They tested lags, and the best-fitted ANN reached a MAPE equal to 26%. Tadano et al. [78] considered the same input and output variables and obtained an ANN with a MAPE of 35%. Finally, Kachba et al. [77] considered CO, NO_x_, O_3_, SO_2_, and PM concentrations and traffic data to infer respiratory hospital admissions and mortality and obtained a MAPE of 28% when analyzing the hospital admissions and 34% when analyzing the mortality.

Our model counted more input variables and a larger database (daily measures in a 12-year period), which could have improved the accuracy of the model resulting in a lower MAPE (11–13%). In addition, we considered data from several air quality monitoring stations, which could better represent the population exposure. According to Seo et al. (2022), by using data from multiple monitoring stations, the error associated with the ANN model can be lowered [79]. The previous studies conducted in São Paulo highlighted the great variability among the data, which may compromise the ability of the model to estimate the most extreme values [49,50]. In this work, the database was split into two seasons with specific meteorological characteristics (Models II and III) and proved to be an efficient way to deal with extreme values, reducing the MAPE to 11% (see Figure 8). The MAPE obtained in the present study is also low in comparison to the ones found in studies conducted in other cities. Polezer et al. [80], for instance, fitted PM_2.5_ concentration, temperature, and relative humidity data in order to estimate respiratory hospital admissions. Their best-fitted ANN resulted in a MAPE of 29%.

Another factor that may have contributed to the results seen in this study is the focus of the analysis on the elderly population, which is the most affected by air pollutants. 

The analysis of the most important variables in each of the models confirms that the data split into two seasons (rainy and dry), generating two distinct models, is a relevant strategy for estimating cardiorespiratory mortality as a function of the considered variables. In Model II, we considered the rainy period as our database. During this period, the relative humidity averaged 80% and, in just 1% of the days, it was below 60%. Keeping in mined indices considered healthy according to the WHO, the importance of this variable for the model was relatively low (6.10%). In other words, it can be inferred that, as the relative humidity remained in an interval considered healthy for almost the entire time series, it had little responsibility for the deaths that occurred on those days.

On the other hand, during the dry season, although the average relative humidity was similar to that of the rainy season (79%), 4% of the days recorded relative humidity below 60%. The greatest variance in the data is the result of days in which the relative humidity was much lower than that recommended by the WHO. In September, for example, relative humidities below 40% were recorded. The low relative humidity of the air, added to colder days, causes the drying of the respiratory surface mucosa, which can facilitate infections [81]. In addition, it decreases the activity of cilia that help remove particles from the air [82]. This impact of low relative humidity on health was captured by Model III, since in this model relative humidity reached an importance almost three times greater in relation to mortality among the elderly (16.67%).

A similar analysis can be performed regarding temperature. In São Paulo, summer occurs during the rainy season, and therefore, this season tends to register higher averages and higher temperature peaks. During the analyzed rainy season, 8.1% of the days had an average daily temperature above 25 °C. On those days, the temperature exceeded 30 °C, setting up possible heat waves. Studies have shown that the elderly are part of the group most affected by the occurrence of heat waves. This is because, with aging, responses to environmental conditions (which include thermoregulation) deteriorate [83]. According to the WHO, high temperatures directly contribute to cardiorespiratory mortality, especially among the elderly. In São Paulo, heat waves have already been associated with excess deaths related to both cardiovascular and respiratory systems. The cardiovascular system seems to be the most affected, and it is estimated that between 1985 and 2005, there were 6% excess deaths among elderly women and 2.5% among elderly men due to the occurrence of heat waves in São Paulo [84]. As in the rainy season, the temperature tends to be higher, and our Model II was able to detect the greater influence of temperature on cardiorespiratory mortality among the elderly. In this case, the importance was 13.1%, while during the dry period (Model III), when temperatures tend to be lower, the importance of temperature was lower (9.5%). According to Nguyen et al. [82], the high temperature associated with high humidity (which is very characteristic of the rainy season in São Paulo) requires an increase in the body’s heat loss through the body surface. This is achieved through blood circulation, resulting in water loss and increased hemoconcentration. This phenomenon overloads the cardiovascular and respiratory systems, and may result in death [82].

In models II and III, PM_10_ and NO_2_ pollutants are among the three most influential variables on elderly cardiorespiratory mortality. The annual mean concentrations registered for both pollutants between 2007 and 2019 were consistently higher than the WHO recommendations. For PM_10_, the annual concentration ranged from 28.35 µg/m^3^ to 41.49 µg/m^3^, while the WHO recommendation is to not exceed 15 µg/m^3^ [52]. For NO_2_, the concentrations ranged from 38.25 µg/m^3^ to 58.36 µg/m^3^, while the WHO recommendation is below 10 µg/m^3^ [52]. Considering the 24 h period, the WHO recommendation is to not exceed 45 µg/m^3^ for PM_10_ and 25 µg/m^3^ for NO_2_ [52]. These values were surpassed in 19.8% of the days for PM_10_ and in 92.4% of the days for NO_2_.

NO_2_ concentration is of concern because is the most critical in relation to WHO recommendations, even though it is not the most important variable according to Models I and II. In São Paulo, vehicles are responsible for 65% of the NO_2_ emissions and, despite the technological advances that have an effect on reducing emissions, the increase in the fleet and the high circulation of old vehicles are obstacles to improving air quality [58,61,85]. According to recent studies focused on limitations imposed on heavy-duty vehicles, these measures may be essential to maintain the NO_2_ concentration within the standards recommended by WHO in São Paulo [66,86].

Exposure to NO_2_ has already been associated with several respiratory symptoms, such as bronchoconstriction, increased bronchial reactivity, airway inflammation, and decreases in immune defense leading to enhanced susceptibility to respiratory infections [87]. All of these are exacerbated in susceptible populations, such as the elderly. According to Larrieu et al. [18], in a cohort study conducted in France between 200 and 2006, the 10 µg/m^3^ enhancement in NO_2_ concentration led to an excess relative risk of 12.3% for upper respiratory disease and 9% for lower respiratory disease among the elderly. The relative risk increase was higher for NO_2_ than for PM_10_ and O_3_ pollutants [18]. In addition to the direct health impacts from NO_2_, it is important to emphasize that NO_2_ concentration is closely related to the concentration of other pollutants, such as PM_2.5_, O_3_, and CO. So, it is unclear to what extent the impacts are due to only to NO_2_ exposure or to the other pollutants too [87].

SO_2_ was the pollutant whose reduction was the most significant in the last 30 years in São Paulo [58]. This reduction was mainly a consequence of the regulation of the stationary sources and of the sulfur content in diesel fuel [58]. The atmospheric concentration of this pollutant remained below the limits recommended by the WHO throughout the time series analyzed in this study. Even so, SO_2_ concentration was identified as having high importance in the cardiorespiratory mortality of elderly people in São Paulo. This result reinforces the need for public policies aimed at reducing the concentration of all pollutants and not just PM_2.5_. Moreover, considering the increase in the global proportion of elderly people among the population, understanding how small changes in air quality and weather conditions affect this group is fundamental to improving the quality of life [11,12].

The comparison between our ANN models with traditional GLMs revealed similar errors, validating the efficiency of ANN in modeling this type of data. It was also possible to observe (see Figure 9 and Figure 10) that the distribution of errors in models constructed from ANNs is less variable than the distribution in GLMs. The superiority of neural networks in relation to other models has also been described by Araújo et al. In the case study carried out by the authors in São Paulo, the neural network presented a MAPE equal to 26%, while the MAPE of the GLM was 35% [49]. It is also important to emphasize that, unlike the most traditional model, an ANN is not subjected to multicollinearity issues. In our GLM models, evidence of multicollinearity is observed, such as negative *β* values describing relationships that should be directly related. In the ANN models, this problem is not present. According to Veaux and Ungar [88], due to the overparameterized aspect of the ANNs, they are insensitive to this problem and they can deal with multicollinear inputs without loss of precision.

Splitting the database into two according to seasons proved to be an efficient strategy in reducing errors for models built from ANNs and GLMs. The error difference found between Model I and Models II and II is probably because of the size of our database (*n* = 4748). When using the whole time series, the data pattern becomes too confusing for ANN to establish a pattern. When it is divided into the rainy and dry seasons, the pattern of the relationship between the variables becomes more constant.

As a gap that may be covered by further research, we suggest lag analyses. In this study, the efforts were focused on exploring the use of ANNs, which are still extremely underexplored in relating meteorological and pollution data to predict health outcomes. We also innovated by proposing the separation of the database as a strategy to reduce model error. Our analysis was limited to lag0, and it is known that although several physiological effects due to pollutants’ presence in the organism occur within hours of exposure, death (especially cardiovascular) may take longer to occur [14,23]. The limitation of the study to lag 0 restricts but does not invalidate the analysis since mortality at lag 0 for both cardiovascular and respiratory causes has also been described by other authors [89,90,91].

As an artificial intelligence (AI) method, the use of ANNs has great potential to face typical challenges from low and middle-income countries, contributing to the achievement of health-related sustainable development goals [92]. Despite this potential, when it comes to the air pollution issue, most approaches have been aiming to forecast human exposure through air quality modeling [47]. Our work demonstrates that ANNs can be used to solve the problem inherent in common models used in multipollutant analysis and that the associated error can be very low.

## 5. Conclusions

This research took advantage of an artificial neural network approach to estimate elderly mortality due to cardiorespiratory issues as a function of the interaction of several air pollutants and meteorological variables. The method proved efficient in dealing with the classical model limitations and resulted in a MAPE of 13%, which is lower than the MAPEs presented in previous studies that applied ANN technics. Additionally, the strategy of splitting the database into two seasons was presented as an efficient strategy to improve the estimation of extreme values and reduce the MAPE. PM_10_ and NO_2_ concentrations were among the most influential variables on cardiorespiratory mortality among the elderly in both models (rainy and dry seasons). The influence of relative humidity is higher during the dry season, and the influence of temperature is higher during the rainy season. Our results support a better understanding of the effect of the combination of air pollutants on human health. In addition, the resulting models have a low associated error, configuring a powerful tool for decision-makers in the evaluation of environmental public policies aimed at public health.

## Figures and Tables

**Figure 1 ijerph-20-05458-f001:**
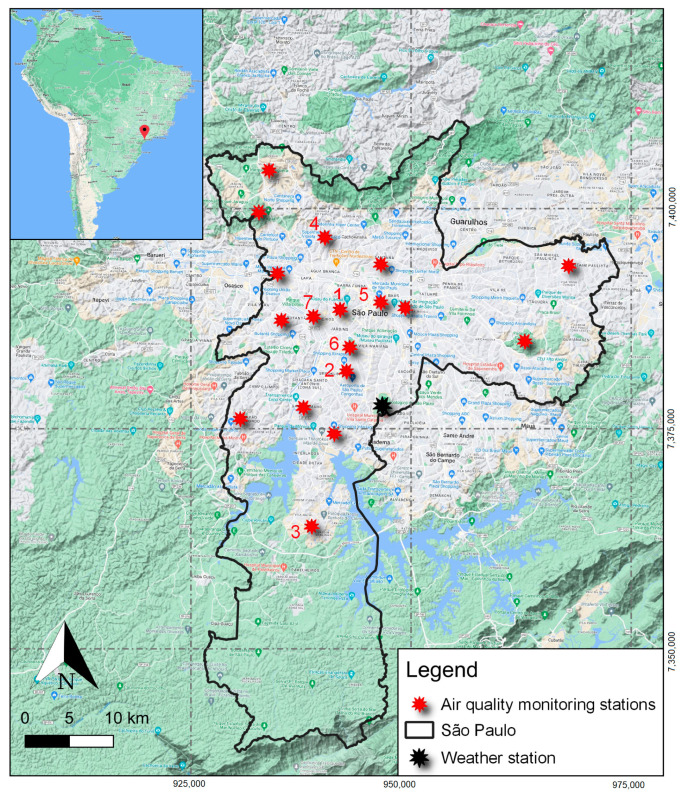
Map illustrating the Sao Paulo municipality and its 18 air quality monitoring stations. The numbers 1 to 7 indicate the following stations: Cerqueira César, Congonhas, Grajaú, Nossa Senhora do Ó, Parque D. Pedro II, Ibirapuera, and Pinheiros. Source: Elaborated by the authors on QGIS 2.18.20 software.

**Figure 2 ijerph-20-05458-f002:**
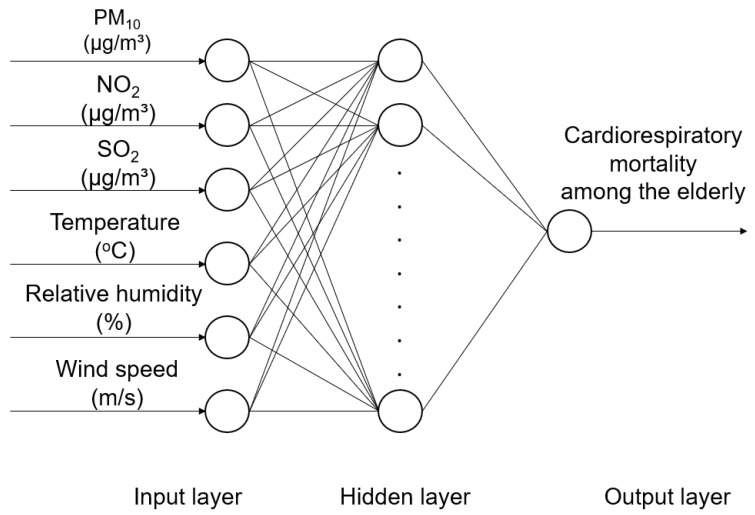
Schematic representation of ANN model with one hidden layer.

**Figure 3 ijerph-20-05458-f003:**
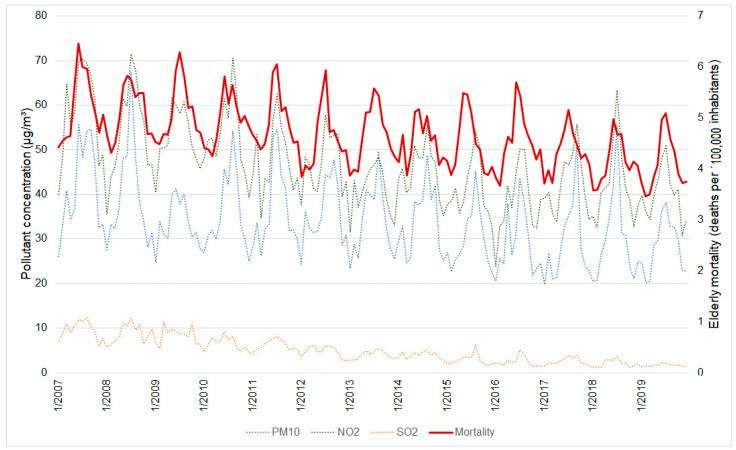
Line graph showing the daily variation of the concentrations of PM_10_, NO_2_, and SO_2_ variables and elderly mortality between 2007 and 2019.

**Figure 4 ijerph-20-05458-f004:**
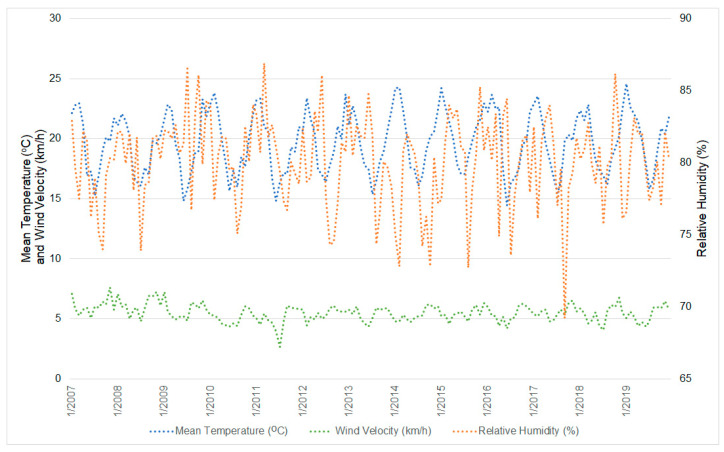
Line graph showing the daily variation of the mean temperature, wind velocity, and relative humidity variables between 2007 and 2019.

**Figure 5 ijerph-20-05458-f005:**
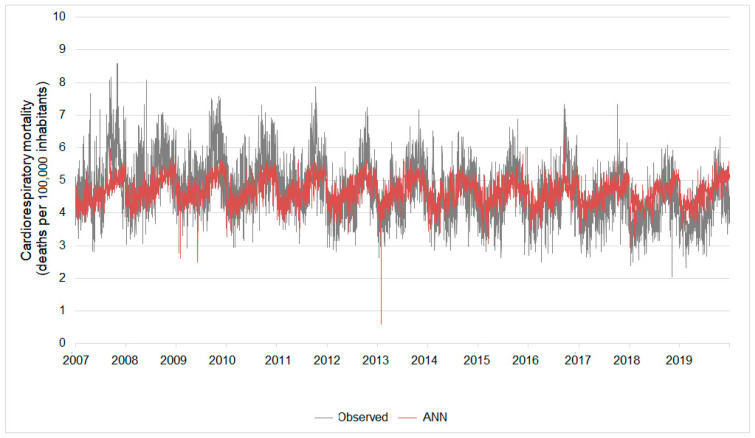
Real daily cardiorespiratory mortality versus that estimated by the Model I ANN. Results considering the ANN build from the entire database (considering rainy and dry seasons).

**Figure 6 ijerph-20-05458-f006:**
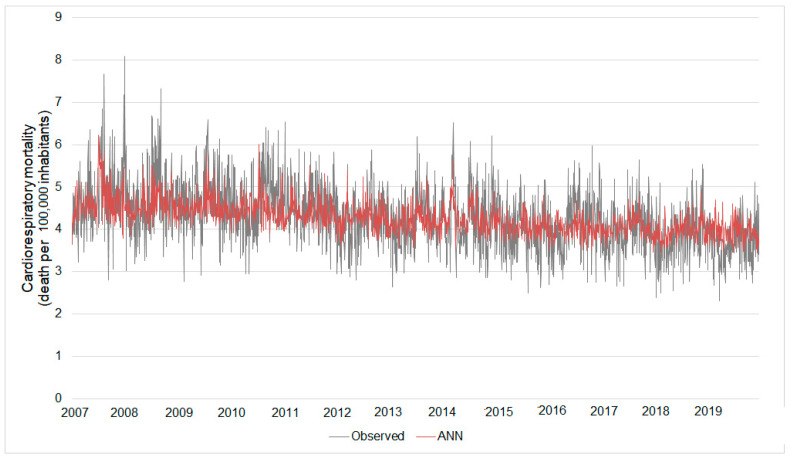
Real daily cardiorespiratory mortality versus that estimated by the Model II ANN. Results considering the ANN build from the rainy season database.

**Figure 7 ijerph-20-05458-f007:**
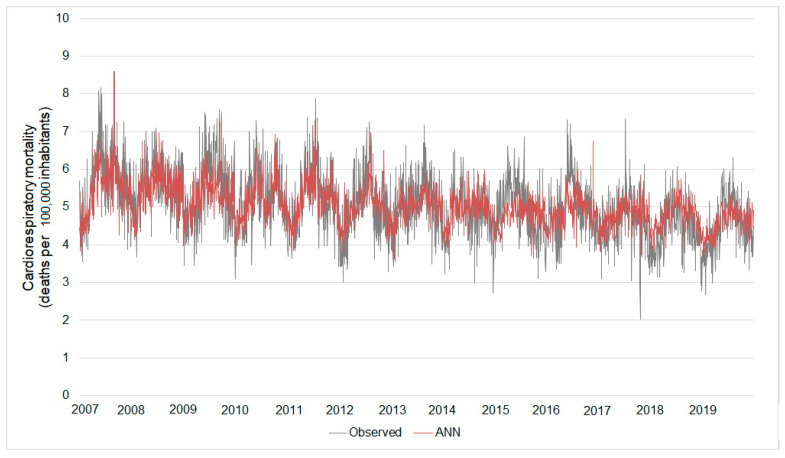
Real daily cardiorespiratory mortality versus that estimated by the Model III ANN. Results considering the ANN build from the dry season database.

**Figure 8 ijerph-20-05458-f008:**
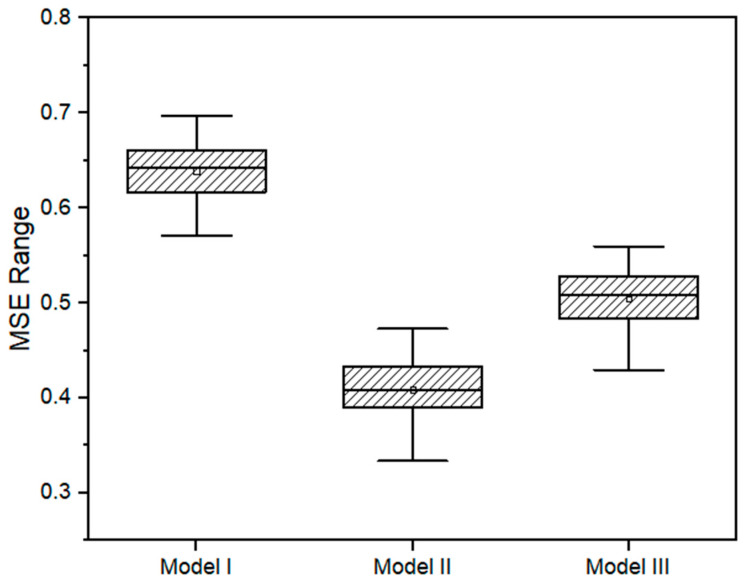
Boxplot indicating the MSE range from the training of the best ANN structure for each model. Vertical bars indicate the 1st and 4th quartiles. The dot inside the box indicates the mean value, and the horizontal bar indicates the median value.

**Figure 9 ijerph-20-05458-f009:**
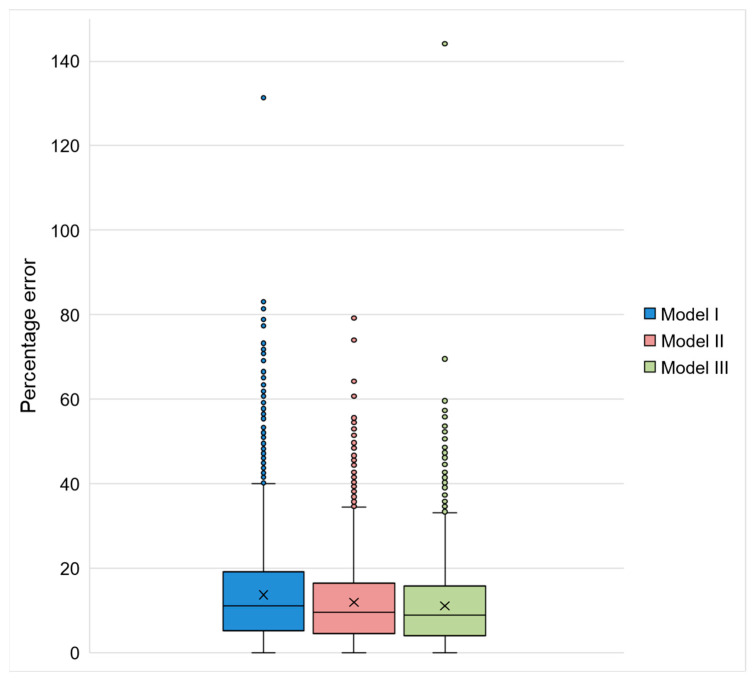
Boxplot indicating the MAPE distribution in Models I, II, and III. The box represents data between the 2nd and 3rd quartiles. Vertical bars indicate the 1st and 4th quartiles, and the circles represent the outliers. The × inside the box indicates the mean value, and the horizontal bar indicates the median value.

**Figure 10 ijerph-20-05458-f010:**
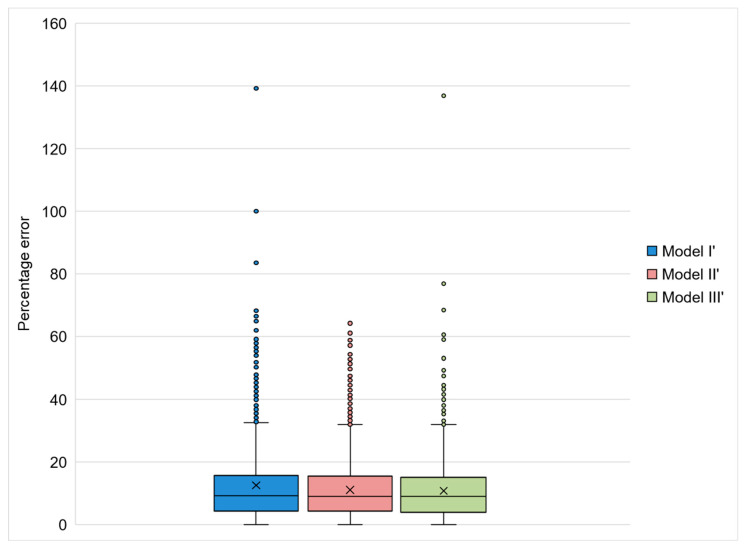
Boxplot indicating the MAPE distribution in Models I’, II’, and III’. The box represents data between the 2nd and 3rd quartiles. Vertical bars indicate the 1st and 4th quartiles, and the circles represent the outliers. The × inside the box indicates the mean value, and the horizontal bar indicates the median value.

**Table 1 ijerph-20-05458-t001:** Relative importance (%) of each variable for cardiorespiratory mortality. The importance is presented according to the three ANNs obtained: the first one considers all databases (Model I), the second considers only data from the rainy season (Model II), and the third considers the data from the dry season (Model III).

	Importance (Position of Importance) (%)		
Variable	Model I	Model II	Model III
PM_10_ concentration	12.32 (4)	68.34 (1)	29.07 (1)
NO_2_ concentration	13.22 (3)	6.81 (3)	27.02 (2)
SO_2_ concentration	46.75 (1)	3.01 (5)	16.51 (4)
Temperature	19.11 (2)	13.10 (2)	9.56 (5)
Relative humidity	4.80 (5)	6.10 (4)	16.67 (3)
Wind speed	3.80 (6)	2.64 (6)	1.17 (6)

**Table 2 ijerph-20-05458-t002:** Estimated parameters for Models I’, II’, and III’. The generalized linear models were elaborated on SPSS software considering gamma distribution and a loglink equation.

	Model I’				Model II’				Model III’			
	*β*	95% CI		*p*-Value	*β*	95% CI		*p*-Value	*β*	95% CI		*p*-Value
(Intercept)	1.626	1.528	1.724	<0.000	1.168	1.021	1.315	<0.000	1.583	1.464	1.703	<0.000
PM_10_	0.000	0.000	0.001	0.500	0.000	−0.001	0.001	0.753	0.001	0.000	0.002	0.008
NO_2_	0.001	0.000	0.001	0.014	0.000	0.000	0.001	0.543	0.001	0.000	0.002	0.036
SO_2_	−0.003	−0.005	−0.001	0.003	−0.002	−0.005	0.000	0.104	−0.003	−0.006	0.000	0.035
Temperature	0.001	−0.002	0.003	0.648	0.011	0.007	0.015	<0.000	−0.008	−0.011	−0.005	<0.000
Relative humidity	−0.002	−0.003	−0.001	<0.000	−0.001	−0.002	0.000	0.009	−0.001	−0.002	0.000	0.035
Wind speed	−0.001	−0.004	0.002	0.570					0.002	−0.002	0.006	0.349

## Data Availability

No new data were created or analyzed in this study. Data sharing is not applicable to this article.

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
