# Peer review of "A Multi-Pollutant and Meteorological Analysis of Cardiorespiratory Mortality among the Elderly in São Paulo, Brazil—An Artificial Neural Networks Approach"

_ijerph, 2023, doi:10.3390/ijerph20085458_

Round 1

Reviewer 1 Report (Previous Reviewer 1)

I revised the previous version of the manuscript. The authors improved the quality of the document. However, some of my appointments were not attended.

Line 146: the error still remains:

7- line 124: “ANNs are virtual machines inspired by the human brain’s operation in information processing” → ANNs are not virtual machines, but machine learning algorithms. Also, the inspiration comes from the nervous system of the superior organisms, not just their brain;

The following problems still remains:

15- Line 216: why did the authors limit the number of neurons to 35? Why do not use 40 or 45 neurons?

16- What are the reasons to test 4 training methods if they are local optimizers? Does it make sense?

22- I understand the “number of deaths” is an integer number;

24- Eqs 4 and 5 must be in Section 2;

25- Another dispersion analysis must be done for the multiple independent runs considering, for example, the boxplot;

26- The authors must consider the lags regarding the exposure to the pollutants.

Author Response

I revised the previous version of the manuscript. The authors improved the quality of the document. However, some of my appointments were not attended.

We thank you for your collaboration with our work by reviewing the manuscript. In our resubmission letter, we had already discussed these notes, but we believe that maybe, as a Reviewer, you did not have access to our responses.

Line 146: the error still remains:

 “ANNs are virtual machines inspired by the human brain’s operation in information processing” → ANNs are not virtual machines, but machine learning algorithms. Also, the inspiration comes from the nervous system of the superior organisms, not just their brain;

Thank you for your comment. We update the definition to:

ANNs are learning algorithms inspired by the nervous system of the superior organisms

The following problems still remains:

15- Line 216: why did the authors limit the number of neurons to 35? Why do not use 40 or 45 neurons?

The group experience shows that the increase in the number of neurons does not lead to a better result. It may generate an overfitted model, instead. Also, previous studies showed that it is more important to evaluate the combination of other models’ hyperparameters (activation functions and learning algorithms), using a limited number of neurons.

We have published several studies using this methodology, which lead to good results, as can be seen in:

https://doi.org/10.3390/cli10010009,

https://doi.org/10.1007/s11696-022-02548-8,

https://doi.org/10.1007/s11696-022-02548-8,

https://doi.org/10.1016/j.dche.2022.100064,

https://doi.org/10.1016/j.cherd.2021.03.008, and

https://doi.org/10.1016/j.ces.2020.116324

16- What are the reasons to test 4 training methods if they are local optimizers? Does it make sense?

As we showed in previous studies, testing different learning algorithms, and the random separation of training and validation data, are important to achieve a good result. The main reason that we have included several ANN configurations is to increase the search for the configuration that leads to a better result.

https://doi.org/10.3390/cli10010009,

https://doi.org/10.1007/s11696-022-02548-8,

https://doi.org/10.1007/s11696-022-02548-8,

https://doi.org/10.1016/j.dche.2022.100064,

https://doi.org/10.1016/j.cherd.2021.03.008,

https://doi.org/10.1016/j.ces.2020.116324

22- I understand the “number of deaths” is an integer number;

The number of deaths is an integer number. However, we considered the mortality rate (number of deaths/100,000 elderly inhabitants). This relativization resulted in a decimal number.

We have considered the mortality rate instead of the number of deaths because the elderly population grew 36% during the analyzed period.

In Fig 2, we have substituted “Cardiorespiratory Deaths” for “Cardiorespiratory Mortality among the elderly”

24- Eqs 4 and 5 must be in Section 2;

Equations 4 and 5 are the equations for the best ANN obtained for each model. They show the activation functions and the number of neurons for this model. They are the obtained result and not a general ANN equation.

25- Another dispersion analysis must be done for the multiple independent runs considering, for example, the boxplot;

We do not believe that a dispersion analysis is important here. In ANN studies we search for the best possible model, and this best model is not obtained by an average analysis. The best parameters are not an average of the parameters obtained in multiple independent runs. Therefore, this is not important for the final result.

26- The authors must consider the lags regarding the exposure to the pollutants.

In fact, most studies involving health outcomes resulting from air pollution exposure consider lags. However, as this is a preliminary analysis using a new model, we limited the study to analyze only the effects on the day of exposure. The study was important to demonstrate that the database split into two seasons is efficient in reducing estimation errors. Based on this finding, we suggest, in the Discussion section, that analyses including lags be carried out in the future in order to explore more possibilities of association.

Reviewer 2 Report (New Reviewer)

Reviewer's report ID ijerph-2241151

Title:  A multi-pollutant and meteorological analysis of the elderly cardiorespiratory mortality in São Paulo, Brazil –an artificial 3 neural networks approach

Date: March 12, 2023

 In this study, multilayer perceptron neural networks were evaluated to associate PM10, NO2, and SO2 concentrations, temperature, wind speed, and relative air humidity with cardiorespiratory mortality among the elderly in São Paulo, Brazil. The daily data from 2007 to 2019 were used. In the manuscript, the results of the GLM model created for mortality were also presented. I have some comments about the models and the interpretation of results.

The comments about the manuscript:

1. About the GLM model. If two variables (mortality and air pollution) are affected by another indicators (environmental policies and the health prevention), then correlation between them will be statistically significant. Therefore, in the GLM model for mortality, the annual trend and seasonal variation (not only wet and dry seasons) should be added.

2. The use of artificial neural networks (ANNs) is OK, but this method does not show the associations between environmental variables and the mortality. The effect of air pollution on human health is widely discussed, but the Discussion chapter absent the interpretations of the ANN results – do not present the number of hidden layer and weights Wij. These parameters should be used in the interpretation of results. The GLM models are widely used for risk assessment, and this is their advantage compared to ANNs. These should be discussed.

Conclusion: A major revision is necessary

Author Response

The comments about the manuscript:

  1. About the GLM model. If two variables (mortality and air pollution) are affected by another indicators (environmental policies and the health prevention), then correlation between them will be statistically significant. Therefore, in the GLM model for mortality, the annual trendand seasonal variation (not only wet and dry seasons) should be added.

Thank you for your comment. As suggested by Conceição et al. 2001, we have added, as explanatory variables, the year, the month, and the day of the week in order to control the seasonality and other trends.

  1. The use of artificial neural networks (ANNs) is OK, but this method does not show the associations between environmental variables and the mortality. The effect of air pollution on human health is widely discussed, but the Discussion chapter absent the interpretations of the ANN results – do not present the number of hidden layer and weights Wij. These parameters should be used in the interpretation of results. The GLM models are widely used for risk assessment, and this is their advantage compared to ANNs. These should be discussed.

Thank you for your comment.  The main objective to use ANNs is to obtain a model in which it is not necessary to know the associations between the inputs and outputs. We just wanted to find a model capable to estimate with sufficient accuracy the mortality based on those six input variables and we believe that we accomplished this task. The best interpretation that ANN results could provide is the degree of importance of each input variable, which we have presented in the Results and in the Discussion sections.

The reviewer can find the number of layers at the beginning of Section 3.2 and the values of the weights and bias of each ANN model were added in a Supplemental File. Finally, a comparison of the advantages of ANNs and GLM models was added in the Discussion section, as suggested by the reviewer.

Round 2

Reviewer 1 Report (Previous Reviewer 1)

The author increase the quality of the manuscript. However, there is a crucial problem regarding the following item:

25- Another dispersion analysis must be done for the multiple independent runs considering, for example, the boxplot;

We do not believe that a dispersion analysis is important here. In ANN studies we search for the best possible model, and this best model is not obtained by an average analysis. The best parameters are not an average of the parameters obtained in multiple independent runs. Therefore, this is not important for the final result.

Unfortunately, this is mandatory. After the user defines the parameters (number of neurons and activation functions), the ANN must be run, but the initialization of the weights is performed at random. As widely known, the cost function based on the MSE or MAE is multimodal for practical problems. In this sense, if the user runs the ANN many times, the final result is not the same. This is directly related to question #16: most training methods are local optimizers, which means the optimizer converges to a local minimum of the basins of attraction it was initialized. This is crucial for all researchers in the ANN field. Note that the authors trained the ANN 50 times. So, I am just asking the boxplot of the dispersion of this 50 simulations, considering the test set. It will increase the quality of the presentation and give the reader an idea of the model's predictability.

Author Response

Thank you again for your availability to discuss our article with us. We are sure that this process enhances the quality of our work.

After your comment, we added the boxplot as suggested. The changes are yellow highlighted in the manuscript.

Reviewer 2 Report (New Reviewer)

The authors provided clarifications and made corrections based on the reviewer's comments.

In Table 2 and in the results chapter, p<0.001 should be instead p=0.000.

Author Response

We thank you for the comments, they have contributed to the better quality of our article.

Reggarding the last comment (In Table 2 and in the results chapter, p<0.001 should be instead p=0.000.), we did the necessary corrections (highlighted in yellow).

Thank you again.

This manuscript is a resubmission of an earlier submission. The following is a list of the peer review reports and author responses from that submission.

Round 1

Reviewer 1 Report

The manuscript proposes using MLPs to evaluate the impact of some pollutants on the cardiorespiratory mortality of elderly people.

Some remarks:

1- Line 31: Burnett et al. (2018) → Burnett et al. [3];

2- Line 51: there are many studies involving tropical countries, especially in Brazil;

3- line 58: “According to a 2016 study” → this information is from 6 years ago;

4- line 60: “The vehicles are mainly responsible for the high concentration of pollutants in the atmosphere” → how about the industries? São Paulo presents a large number of industries. This is also a great problem ;

5- line 71: Bravo (2015) → Bravo et al. [31]. Please correct all over the manuscript. Do not write the year, just the citation;

6- Ref 45 is wrong. There is no 7th edition of this book;

7- line 124: “ANNs are virtual machines inspired by the human brain’s operation in information processing” → ANNs are not virtual machines, but machine learning algorithms. Also, the inspiration comes from the nervous system of the superior organisms, not just their brain;

8- Line 128: there are neural architectures that uses less than 3 layers, such as the ELM. Also, other ones do not present layers. This statement is related to just some ANNs;

9- Line 129: “trained to predict outputs” → it depends on the task to be solved. In the case presented in this study, the authors are performing nonlinear mapping, not a prediction. Therefore, they are estimating the outputs. The term “prediction” is directly related to time series forecasting;

10. Line 149: which ANN will be used? ANNs is a class of models, not a specific one;

11- The authors must explain what mean the circles and lines in Figure 2;

12- Line 207: Equation 1 must be nominated in the text. The same for the other equations, tables, and figures;

13- Line 209: the variables must be in italics, and the subscribed must be observed;

14- Equation 1 is wrong. In an MLP, it is necessary to specify the activation function (usually a nonlinear function), and the explanation is related just to the input layer. For example, if were are dealing with the 1st hidden layer, xi is not an input variable but the output of the previous layer;

15- Line 216: why did the authors limit the number of neurons to 35? Why do not use 40 or 45 neurons?

16- What are the reasons to test 4 training methods if they are local optimizers? Does it make sense?

17- The same activation functions were used in all layers? 

18- There is no reason to use a linear function in nonlinear mapping tasks;

19- If the authors would like to perform a comparative study on the different versions of the MLP, the test set must always be the same;

20- I have concerns about the definition of the variables' importance since the ANN is a nonlinear mapping. The importance could vary depending on the initialization because the cost function based on the MSE presents many local optimum points;

21- It is difficult to see the lines in Figs 3 and 4;

22- I understand the “number of deaths” is an integer number;

23- Section 3.2: it is unclear how many independent runs the authors performed with each combination of parameters. If they run the MLPs just once, the results are invalid;

24- Eqs 4 and 5 must be in Section 2;

25- Another dispersion analysis must be done for the multiple independent runs considering, for example, the boxplot;

26- The authors must consider the lags regarding the exposure to the pollutants;

27- Line 394: the authors aim to compare their results with other studies. However, their database, output target, and size of the input variables are not the same. These results are not comparable. 

28- The manuscript does not present a comparison among other models. The authors just use the MLP with small variations;

The manuscript does not present any novelty. Indeed it is just a case study that follows the steps developed by other investigations. This is not a problem. However, the application of the tool presents many problems, as well as their presentation. The most important errors are the absence of comparison with other models and the absence of more than 1 independent run for each topology of MLP. Finally, the authors must be careful when comparing the results of other investigations since they are using other databases. They indicate 4 other studies. I read all of them and identified many differences. For example, the outputs regarding the number of hospitalizations of the entire population is a different target from the deaths of elderly people. 

Reviewer 2 Report

Dear authors

The problem addressed in this research is interesting and your focus in data coming from countries with a tropical climate represents an interesting point.

Your research is clearly presented and your article is well written and clear. It is well written and the results and the method used to generate the model is clear. The results are presented in an honest and clear way and this is remarkable nowadays.

However, I have strong concerns about the novelty and usefulness of the research, linked to the methodological proposal and interpretation of the results.

Main concerns about methodological issues:

1) Validity of the method.The authors claim that the main novelty resides in the use of a multilayer perceptron neural networks to associate PM10, NO2, and SO2 concentrations, temperature, wind speed, and relative air humidity with cardiorespiratory mortality among the elderly in São Paulo, Brazil. However, there are many previous research (some of them commented in the article) with this approach, the so claimed novelty lies in the introduction of meteorological variables. However, the introduction of more explanatory variables (input data) in the ANN which are strongly correlated with the previous one (temperature, wind speed, and relative air humidity are strongly correlated with PM10, NO2, and SO2 concentrations) may lead to redundancy and loss of efficiency in the ANN model. So, these structures result in correlated inner representations abstracted by the network and thus in poor results in new scenarios for the goal of mortality prediction. In these cases, several problems could arise:

·       a local minimum of the cost function can be reached, and so the network isn't complex enough to get a better approximation of the function. Network   accuracy to  approximate any given function is then compromised.  

·       The learning rate becomes too large. The model goes too far in the direction of the gradient and keeps overshooting the target.

·       Numerical instability.  

·       The model may converge, but to the wrong parameters.

·       Neural networks are not right for every task and so the results can not be extrapolated to other cities or scenarios

·       Overfitting because the network learns from noise in the correlated data (not necessary linked to correlation itselft)

In this case, authors should prove the validity of the method, answering to those issues to prove the chosen method is reliable, and the opportunity to include meteorological inputs in the model becomes useful and trusty.

2)  Regarding hypotheses and framework of the model. In the introduction, authors comment about the multipollutant models. It is well known that the results on the interaction of pollutants and their health effects are contradictory. While some authors comment on additive effects, others describe independent effects. In the article it is referred that Santana et al. (2020) demonstrated the positive relationship between pollutants  PM, O3, CO, SO2, and NO2 (individually analyzed) and hospitalizations due to respiratory 77 diseases [33]. A deep discussion on the interaction model or hypotheses considering these facts (interaction of pollutants and health effects) would be necessary to establish the framework of the proposed model. There is a need on including the foundations of ANN related models and their inner relationship based on muti-pollutant mixture exposure interactions as the main hypotheses of the proposed approach.

Main concerns about results:

1) Introduction of seasonality in the models seems to be important but it could be unneccessary to include meteorological variables. This introduction hampers the neural networks to accurately predict the highest and lowest values, and maybe that better results can be achieved if authors use pollutants PM, O3, CO, SO2, and NO2 as a explanatory variables. As it is shown in the article and the bibliography, the most influential variables in cardiorespiratory mortality among the elderly were PM10 and NO2 concentrations, and the use of more variables is not demonstrated in the research. In fact, the multipollutant analysis from Miranda et al. (2021) got better results than the proposal in a similar scenario and data. Perhaps the results should concentrate on exploring this seasonality with the pollutant variables.  

2) More information regarding the link between temperature, humidity episodes and pollutant concentration data registered in Sao Paulo would help to explain some results commented in the article. In fact, as it is written in the article, the following paragraph  support the previous comment of the reviewer: “The annual variation in the concentration of pollutants is a consequence of the meteorological   characteristics typically found in the two seasons of the year (rainy and dry), being the  months between May and September the most unfavorable to the pollutants dispersion [53]. There is an increase in the  concentration of particulate matter in the Metropolitan Region of São Paulo, on days with   high temperature, low relative humidity, absence of precipitation, and weak ventilation   [66]. Considering this, the worst pollution days tend to occur during the winter (between   June and September), when these factors are more frequently associated [65]. In São Paulo,  the winter is characterized by the driest period of the year. Thus, in addition to the greater  amount of pollutants that penetrate the respiratory system, at this season, the elderly are   also more susceptible to infections such as influenza and pneumonia, which are common   during the winter and contribute to cardiorespiratory impairment [67,68].”

3) It is found that the biggest errors were concerning  the predictions of the lowest values of cardiorespiratory mortality (Figure 7). The model tends to overestimate the mortality. Some explanation would be interesting.

4) Previous ANNs that related human health outcomes to pollutant concentrations in   the São Paulo municipality reached higher MAPEs than the ones found in this work. Thus, it is not clear the advantage of this proposal.

5) Although physiological effects due to pollutants’ presence in the organism occur within hours of exposure, death (especially cardiovascular), takes longer to occur [14,22]. This fact could invalidate or render unhelpful the proposed approach. Authors should justify this issue.

Some other minor concerns:

1) More information regarding the location of the monitoring stations from CETESB database and the meteorological stations where daily means of air temperature, relative humidity, and wind speed are measured will be of interest to show the link between them. Several air qualitymonitoring stations are used but the authors should justify why they could better represent the population exposure than other data used in previous research.

2) Please, provide the size of the datasets and explain the size of the set used for training, validation and prediction in the ANN in section 2.2

 3) The influence of relative humidity is higher during the dry season and the influence of the temperature is higher during the rainy season. Some explanation could help the reader.
